# Abnormal Cockpit Pilot Driving Behavior Detection Using YOLOv4 Fused Attention Mechanism

**Nongtian Chen [1,*], Yongzheng Man [2] and Youchao Sun [3]**

1   College of Aviation Engineering, Civil Aviation Flight University of China, Guanghan 618307, China
2   College of Civil Aviation Safety Engineering, Civil Aviation Flight University of China,
    Guanghan 618307, China
3   College of Civil Aviation, Nanjing University of Aeronautics and Astronautics, Nanjing 211106, China
*   Correspondence: chennongtian@hotmail.com; Tel.: +86-83-8518-3621

**Abstract:** The abnormal behavior of cockpit pilots during the manipulation process is an important incentive for flight safety, but the complex cockpit environment limits the detection accuracy, with problems such as false detection, missed detection, and insufficient feature extraction capability. This article proposes a method of abnormal pilot driving behavior detection based on the improved YOLOv4 deep learning algorithm and by integrating an attention mechanism. Firstly, the semantic image features are extracted by running the deep neural network structure to complete the image and video recognition of pilot driving behavior. Secondly, the CBAM attention mechanism is introduced into the neural network to solve the problem of gradient disappearance during training. The CBAM mechanism includes both channel and spatial attention processes, meaning the feature extraction capability of the network can be improved. Finally, the features are extracted through the convolutional neural network to monitor the abnormal driving behavior of pilots and for example verification. The conclusion shows that the deep learning algorithm based on the improved YOLOv4 method is practical and feasible for the monitoring of the abnormal driving behavior of pilots during the flight maneuvering phase. The experimental results show that the improved YOLOv4 recognition rate is significantly higher than the unimproved algorithm, and the calling phase has a mAP of 87.35%, an accuracy of 75.76%, and a recall of 87.36%. The smoking phase has a mAP of 87.35%, an accuracy of 85.54%, and a recall of 85.54%. The conclusion shows that the deep learning algorithm based on the improved YOLOv4 method is practical and feasible for the monitoring of the abnormal driving behavior of pilots in the flight maneuvering phase. This method can quickly and accurately identify the abnormal behavior of pilots, providing an important theoretical reference for abnormal behavior detection and risk management.

**Keywords:** pilot abnormal behavior; behavior detection; YOLOv4 algorithm; CBAM; flight safety

## 1. Introduction

Overall, 60% to 80% of flight accidents are caused by human factors. The statistics from the Civil Aviation Safety Annual Report show that in the past 10 years, the proportion of flight accidents caused by pilot and flight crew factors has been as high as 67.16% [1]. With the rapid development of civil aviation, the air transportation volume has increased significantly, and ensuring aviation safety has resulted in higher requirements for civil aviation pilots. According to relevant aviation accident statistics, most of the flight accidents are caused by the abnormal behavior of pilots, and the abnormal behavior of pilots in the cockpit is directly or indirectly related to flight accidents and symptoms. On 10 July 2018, an oxygen mask incident occurred in the airspace of Guangzhou on a flight from Hong Kong to Dalian in China. The investigation results showed that the cause of the incident was that the co-pilot smoked electronic cigarettes in the cockpit (abnormal behavior). An adjacent air conditioning unit was mistakenly shut down, resulting in a lack of oxygen

in the cabin, triggering the incident. The Civil Aviation Administration of China issued an advisory circular (AC-121-FS-2018-130) on the flight operation style of civil aviation pilots, regulating pilots during the whole process of flight operations from the pre-flight stage to the flight operation stage, post-flight, and short-stop and station-based stages, driving behavior to improve the professionalism of pilot teams. Possible solutions for how to identify and monitor the abnormal behavior of pilots effectively, prevent the possible consequences of the risk of abnormal behavior of pilots, and explore and establish an effective mechanism to reduce human errors from the perspective of intrinsic safety have attracted the attention of many researchers. Therefore, it is of great practical significance to carry out research on the identification, monitoring, and early warnings of the abnormal driving behavior of pilots to regulate this driving behavior of pilots and ensure the safety of aviation operations.

Abnormal behavior research originated in the 1960s, and first explored the mechanism of abnormal behavior from the perspective of the behavioral environment [2]. Action recognition is an important field in computer vision and has been the subject of extensive research. It is widely used in pedestrian detection [3], robot vision [4], car driver detection [5], intelligent monitoring [6], and worker detection [7]. With the development of information technology, more scholars are using information detection technology to carry out abnormal behavior research. Abnormal behavior identification and detection processes are used to locate and detect abnormal actions; that is, the accurate identification of a certain action. The traditional detection technology has problems such as poor robustness to changing targets and long and redundant detection windows, which limit the improvement of the accuracy and speed during target detection. With the emergence of convolutional neural networks, due to their better representative learning ability, the research on abnormal behavior detection began to develop in the direction of convolutional neural network technology. In 2017, Wang et al. [8] tried to use the depth map method for the first time to identify the hand movements of cockpit pilots and to implement an approach and landing safety analysis. Liu et al. extracted time series of 3D human skeleton key points using Yolov4 and applied a mean shift target tracking algorithm, then converted key points into spatial RGB data and put them into a multi-layer convolution neural network for recognition [9]. Zhou et al. proposed a new framework for behavior recognition [10]. In this framework, we propose an object depth estimation algorithm to compute the 3D spatial location object information and use this information as the input to the action recognition model. At the same time, to obtain more spatiotemporal information and better deal with long-term videos, combined with the attention mechanism, spatiotemporal convolution and attention-based LSTMs (ST-CNN and ATT-LSTM) are proposed. Incorporating deep spatial information into each segment, the model focuses on the extraction of key information, which is crucial for improving the behavior recognition performance. Some scholars have proposed an abnormal target detection method based on the T-TINY-YOLO network model. The YOLO network model is used to train the calibrated abnormal behavior data to achieve end-to-end abnormal behavior classification, thereby achieving abnormal target detection for specific application scenarios [11]. Some scholars have studied the impact of civil aviation pilots' work stress on unsafe behavior based on a correlation analysis and multiple regression analysis [12]. In 2018, Yang et al. used the heads-up display to perform pattern recognition for pilot behavior, and for the first time proposed a behavior recognition framework that included pilot eye movements, head movements, and hand movements [13]. The deep-learning-based anomaly detection reduces human labor and its decision making ability is comparatively reliable, thereby ensuring public safety. Waseem et al. proposed a two-stream neural network in this direction for anomaly detection in surveillance [14]. Qu proposed a future frame prediction framework and a multiple instance learning (MIL) framework by leveraging attention schemes to learn anomalies [15]. Other scholars have used 3D ConvNets to identify anomalies from surveillance videos [16]. Waseem et al. presented an efficient light-weight convolutional neural network (CNN)-based anomaly recognition framework that is functional in surveillance environments with reduced time

complexity [17]. One-shot image recognition has been explored for many applications in the computer vision community. One-shot anomaly recognition can be efficiently handled according to the 3D-CNN model [18]. The low reliability during feature and tracking box detection is still a problem in visual object tracking, An et al. proposed a robust tracking method for unmanned aerial vehicles (UAV) using dynamic feature weight selection [19]. Wu designed a road travel time calculation method across time periods. Considering the time-varying vehicle speed, fuel consumption, carbon emissions, and customer time window, the satisfaction measure function and economic cost measure function based on the time window were adopted [20]. Regarding the study by Chen [21], in order to improve the accuracy and generalization ability during hyperspectral image classification, in their paper a feature extraction method combining a principal component analysis (PCA) and local binary pattern (LBP) was developed for hyperspectral images, which provided a new idea for processing hyperspectral images. Zhou [22] proposed an ant colony optimization (ACO) algorithm based on parameter adaptation using a particle swarm optimization (PSO) algorithm with global optimization ability, a fuzzy system with fuzzy reasoning ability, and a 3-Opt algorithm with local search ability, namely PF3SACO. Yao et al. proposed a scale-adaptive mathematical morphological spectral entropy (AMMSE) approach to improve the scale selection. In support of the proposed method, two properties of the mathematical morphological spectra (MMS), namely the non-negativity and monotonic decrease, were demonstrated [23].

In recent years, deep learning has achieved outstanding performance in many fields, such as image processing, speech recognition, and semantic segmentation. Now, the commonly used neural networks include deep Boltzmann machines (DBM), recurrent neural networks (RNNs) [24], and convolutional neural networks (CNNs) [25]. In 2015, Girshick [26] first proposed the R-CNN algorithm for abnormal behavior recognition, which effectively improved the recognition accuracy. The improved algorithms, such as Fast R-CNN and Faster R-CNN, proposed later have higher efficiency and accuracy in abnormal behavior recognition [27,28]. These improved methods improve the speed of the information collection, information processing ability, and transmission speed, and provide important theoretical and technical support for abnormal behavior identification and early warnings. There are many regional models based on deep learning, including SSP [29], SSD [30], and YOLO [31,32].

In short, many scholars have carried out studies on abnormal behavior recognition and have achieved many effective results, but further research is needed on the abnormal behavior recognition algorithms and monitoring effects, especially as research combined with the abnormal behavior of pilots in the civil aviation industry is rare. This paper proposes an abnormal pilot behavior monitoring and identification algorithm based on an improved YOLOv4 (you only look once) approach, adopts a deep-learning-based abnormal behavior target detection algorithm, and introduces a convolutional attention mechanism module (CBAM) for the feature fusion of the backbone network. A convolutional back attention module is used to enhance the perception of the model in the channel and space, and finally to extract the features through the convolutional neural network and monitor and identify the abnormal behavior of pilots in order to provide a reference for the identification of the abnormal behavior of pilots and the norms of pilot behavior.

## 2. Overview of the Method

### 2.1. Convolutional Neural Networks

In recent years, convolutional neural networks (CNNs) have made great progress in image and video processing. The CNNs extract the high-level semantic features of images through the deep neural network structure, and complete the recognition and classification of complex images and videos. A convolutional neural network is generally a feed-forward neural network formed by overlapping convolutional layers, pooling layers, and fully connected layers, and the characteristics include local connections, weight sharing, and aggregation. These properties make convolutional neural networks invariant to certain

degrees of translation, scaling, and rotation. The role of the convolutional layer is to extract the features of a local area, and the different convolution kernels are equivalent to different feature extractors. The role of the pooling layer is to perform feature selection and reduce the number of features, thereby reducing the number of parameters. The learning rate is a very important parameter in such algorithms. Here, Softmax is selected as the classifier, and the optimization algorithm of the learning rate is the adaptive algorithm Adam. Its calculation formula is:

$$m_t = \beta_1 m_{t-1} + (1 - \beta_1)g_t \tag{1}$$

$$v_t = \beta_2 v_{t-1} + (1 - \beta_2)g_t^2 \tag{2}$$

Here, $t$ is the time, $m_t$ is the first-order moment estimation of the gradient, $v_t$ is the second-order moment estimation of the gradient, and $\beta_1$ and $\beta_2$ are the exponential decay rated of the moment estimation, ranging from 0 to 1. When calculating the deviation correction, Equation (2) will be used, where $\hat{m}_t$ and $\hat{v}_t$ are the corrections of the sum:

$$\hat{m}_t = \frac{m_t}{1 - \beta_1{}^t} \tag{3}$$

$$\hat{v}_t = \frac{v_t}{1 - \beta_2{}^t} \tag{4}$$

The gradient is updated using Equation (5):

$$\theta_{t+1} = \theta_t - \mu \hat{m}_t / (\sqrt{\hat{v}_t} + \varepsilon) \tag{5}$$

Here, $\varepsilon$ is a numerically stable small constant; $\theta_t$ represents the gradient to be updated, generally $10^{-8}$; and $\mu$ is the step size, generally 0.001.

### 2.2. YOLO

The YOLO algorithm is an object recognition and localization algorithm based on a deep neural network. It is characterized by improving the speed of the deep learning target detection process and meeting the requirements for real-time monitoring to a certain extent. The CNN algorithm convolves the image through the convolutional neural network, but the detection speed is low, which cannot meet the needs of real-time monitoring. The characteristic of the YOLO algorithm is that only one CNN operation is needed for the image, and the corresponding region and position of the regression prediction frame can be obtained in the output layer. The algorithm steps are as follows.

Divide the original image into $S \times S$ grid cells. If an object falls in the grid, then the feature of the grid is the object (if multiple objects fall in this grid, the closest object in the center is the feature of this grid).

(1)    The features of each grid need $B$ bounding boxes (Bbox) to return. To ensure accuracy, the $B$ box features corresponding to each grid are also the same.

(2)    Each $B$box also predicts 5 values: $x, y, w, h,$ and the confidence. The $(x, y)$ is the relative position of the center of the $B$box in the corresponding grid, and $(w, h)$ is the length and width of the $B$box relative to the full image. The range of these 4 values is [0, 1], and these 4 values can be calculated from the features. The confidence is a numerical measure of how accurate a prediction is. Let us set the reliability as $C$, where $I$ refers to the intersection ratio between the predicted $B$box and the ground-truth box in the image; the probability of containing objects in the corresponding grid of the $B$box is $P$, and the formula is:

$$C = P i \tag{6}$$

(3)    If the grid corresponding to the $B$box contains objects, then $P = 1$, otherwise it is equal to 0. If there are $N$ prediction categories, plus the confidence of the previous $B$box

prediction, the $S \times S$ grid requires $o$ output information. The calculation method is as follows:

$$o = S \times S \times (1 + B + N) \tag{7}$$

For each grid, the confidence that it belongs to each category will also be predicted. Among them, the $B$ boxes can only belong to one category, which corresponds to the first step, and its characteristics are the same.

### 2.3. YOLOv4

2.3.1. CSPDarkent–53

CSPDarknet-53 is based on the YOLOv4 backbone network, and is modified and improved on the basis of it, finally forming a backbone structure that includes 5 CSP modules. The CSP module divides the feature map of the base layer into two parts; that is, the original stack of residual blocks is split into two different parts on the left and right. The main part is used to continue the original stack of residual blocks, and the other part is similar to the residual edge, which is directly connected to the end after a small amount of processing. They are then merged through a cross-stage hierarchy. Through this processing, the accuracy of the model is also ensured based on reducing the amount of calculation.

2.3.2. Prediction Box Selection

The prediction principle of YOLOv4 is to divide the image into $13 \times 13$, $26 \times 26$, and $52 \times 52$ networks, and each network node is responsible for the prediction of one area. YOLOv4 uses a clustering method to select candidate frames, and the cluster centers are divided into 3 scales of different sizes according to the different sizes used for prediction. The calculation formula for the offset predicted by the network is:

$$b_x = \sigma(t_x) + c_x \tag{8}$$

$$b_y = \sigma(t_y) + c_y \tag{9}$$

$$b_w = p_w e^{t_w} \tag{10}$$

$$b_h = p_h e^{t_w} \tag{11}$$

Here, $(c_x, c_y)$ is the distance from the upper left corner (when the prediction frame is selected, the values of $c_x$ and $c_y$ are 1); $(p_w, p_h)$ are the length and width of the prior frame, respectively; $p_w$ and $p_h$ are determined manually; $(t_x, t_y)$ is the offset of the target center point relative to the upper left corner of the grid, where the prediction point is located; $(t_w, t_h)$ are the width and height of the prediction frame, which are related to $p_w$ and $p_h$, respectively (see Equations (10) and (11)) and with which the width and height of the *B*box are obtained; $\sigma$ is the activation function, indicating the probability between [0, 1].

## 3. The Improved Network Model

### 3.1. Channel Attention Mechanism

An attention mechanism is a method of processing data, which imitates the human visual system, integrates local visual structures, focuses attention on important points among a lot of information, selects key information, and ignores other unimportant information. A Channel attention block (CAB) can model the dependencies of different channel features, fuse multi-channel feature images, and adaptively adjust their feature weights. The channel attention module rescales the weights of each input channel so that the key region feature channels containing the target object have a greater contribution during convolution. The idea is to enhance the weight of the key channels and reduce the weight of invalid channels. The channel attention can be expressed as Equation (12):

$$M_C(F) = \sigma(MLP(AvgPool(F)) + MLP(MaxPool(F))) \tag{12}$$

In the formula, *F* represents the feature of the input, where $\sigma$ represents the activation function, and *AvgPool*() and *MaxPool*() represent the processes of average pooling and maximum pooling, respectively.

### 3.2. Spatial Attention Mechanism

In the process of behavior recognition, when the pilots perform abnormal behaviors such as calling and smoking, the location features (such as gradients and grayscales) will change drastically. Therefore, the spatial attention mechanism (SAB) can be used in the feature map. By increasing the weights of the key parts, the network can focus more on crucial features and improve the feature extraction ability of the network. The channel attention mechanism is the part that the network one must pay attention to, and the spatial attention mechanism gives the locations of key features. The specific implementation process is shown in Equation (13):

$$M_S(F) = \sigma(f^{7\times7}([AvgPool(F); MaxPool(F)]))　　　　　(13)$$

In the formula, *F* also represents the feature of the input; $7 \times 7$ convolution is used for feature extraction, then average pooling and maximum pooling are used for evaluation, and finally normalization is performed according to the activation function $\sigma$.

### 3.3. Attention Mechanism Fusion

This article involves the fusion of the spatial attention mechanism and channel attention mechanism, which will be used for the detection of behavior recognition in the YOLOv4 algorithm. For an input video F, the global information for each feature channel is obtained using the global average pooling and maximum pooling operations, and then the future channel attention vector is obtained through two fully connected layers, which are used to weigh the input feature F channel using $M_C$(F). In addition, this feature is input to the $3 \times 3$ convolution layer and output by the sigmoid function and $M_S$(F), which gives the feature *F'*. Its structure is shown in Figure 1. In this paper, the attention mechanism is added to the two effective feature layers extracted from the backbone network, and the attention mechanism is also added to the results after up-sampling. The attention mechanism in this paper can enhance the feature extraction ability of the model based by increasing a small amount of the computation. Due to the large size gap in the image dataset, after the attention mechanism is introduced, the image features can be extracted from multiple scales, which strengthens the model's ability to detect images.

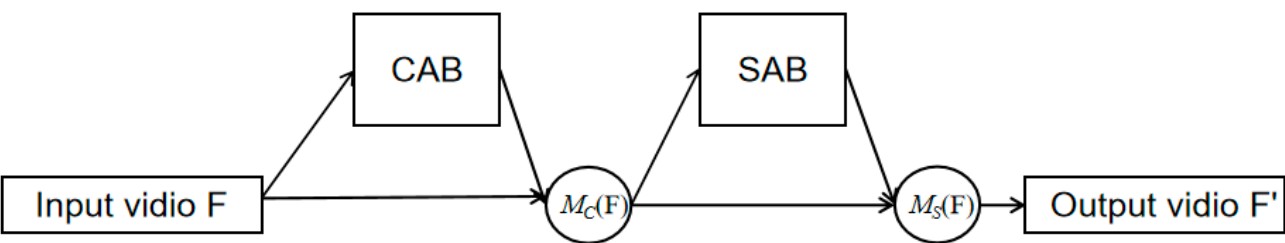

**Figure 1.** Fusion of CAB and SAB attention mechanisms.

## 4. Test and Analysis

### 4.1. Environment Settings

The environmental configuration of this experiment is shown in Table 1, and the hardware configuration of the comparison experiment is the same configuration.

**Table 1.** Numbers of various types of images.

| Operating System | Windows 10 |
|---|---|
| CPU | i7-10750H |
| RAM | 16G |
| GPU | RTX 2060Ti |
| Language | Python 3.6 |
| Backbone Network | CSPDarkent–53 |

The research subjects in this paper are pilots. There is no special public dataset available at present, so the database must be established by itself. The database data in this paper mainly come from the relevant action pictures taken by us, pictures that meet the requirements for the existing datasets, and relevant pictures searched on the Internet. The dataset is prepared according to the deep learning standard dataset format in VOC 2007. The specific steps are: (1) use the labeling tool to classify the abnormal behavior in the image, whereby the category names are calling and smoking; (2) create relevant files according to the standard dataset format and save the files, including pictures, sizes, and coordinates for target detection, then divide the dataset into a training set and test set at a ratio of 9:1. Table 2 shows the numbers of images in the various categories in the dataset.

**Table 2.** Numbers of various types of images.

| Type | Calling | Smoking |
|---|---|---|
| Number | 500 | 500 |

### 4.2. Detection Process

According to the driving behavior requirements and flight guidelines for civil aircraft pilots, smoking and calling during the flight can be called abnormal pilot behaviors. Therefore, in the identification process, these typical abnormal behaviors are identified to provide a basis for the implementation of abnormal driving behavior monitoring and early warning processes. The process of detecting abnormal pilot behaviors is shown in Figure 2. The main process is as follows.

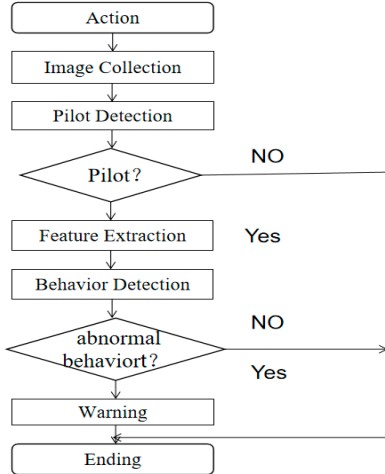

**Figure 2.** Abnormal behavior detection process.

(1) Taking with a camera in the cockpit of an aircraft simulator, capture images in frame units from the live video stream captured by the camera;

(2) Perform abnormal behavior detection on the pilots. Use the model trained by the deep learning YOLO v4 algorithm to locate the pilot area, and when the area is detected, the abnormal behavior can be identified according to the model;

(3)   Monitor the video. When there is abnormal behavior, it will give a warning. After the frame detection ends, enter the next frame.

### 4.3. Model Training

In the improved YOLOv4 model training, the smaller the loss value of the model structure the better, and the expected value is 0. To achieve the best performance for the model, during training the number of iterations is set to 600, the weight decay coefficient is set to 0.0001, and the learning rate momentum is set to 0.9 to prevent the model from overfitting. The maximum training batch is set to 8, the loss function value drops sharply from 0 to 300 times, and the loss number decreases slowly from 300 to 600 times. After 400 iterations, the loss value tends to stabilize around 0.05, and the model reaches the maximum excellence state. The training loss is shown in Figure 3.

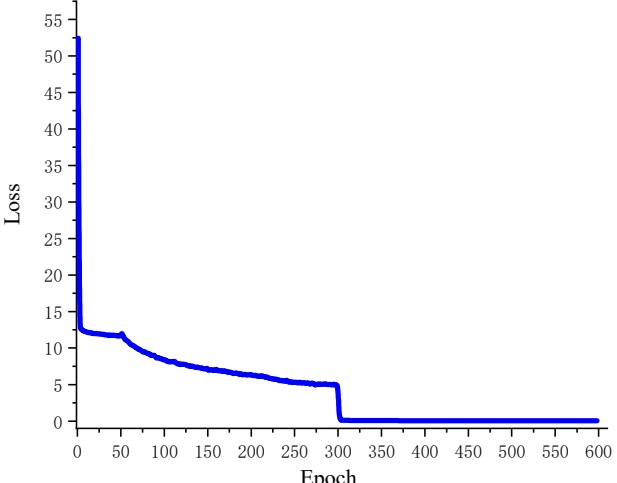

**Figure 3.** Loss map.

### 4.4. Evaluation of the Model Performance

It can be seen from the Figure 4 below that for the two types of abnormal behaviors, the recognition rate of the improved YOLOv4 is significantly higher than that of the unimproved algorithm.

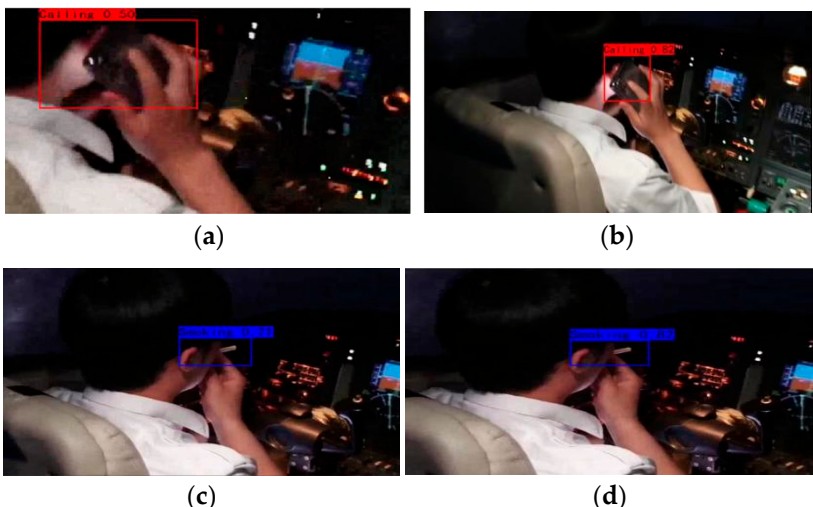

**Figure 4.** Object detection results using the YOLOv4 method (**a**,**c**) and object detection results using our proposed method (**b**,**d**).

When compared with the original YOLOv4 algorithm, the unified video is input and the darknet backbone network is used for training. In order to make the model converge as soon as possible, this experiment adopts the method of transfer learning. The experimental data are shown in Table 3.

**Table 3.** Comparison between the original YOLO algorithm and the improved YOLO algorithm used in this article.

|         | YOLOv4/% | Improved YOLOv4/% |
|---------|----------|-------------------|
| Smoking | 71       | 82                |
| Calling | 50       | 82                |

In the evaluation of the pilots' abnormal behavior recognition effect, the important parameters are as follows: $T_P$ indicates that abnormal behavior is detected, and there is also abnormal behavior in the actual picture (the number of samples detected by the algorithm); $T_N$ indicates that no abnormal behavior is detected, and the actual picture is not abnormal (the number of correct error samples detected by the algorithm); $F_N$ means that no abnormal behavior is detected, but abnormal behavior is present in the actual graph (the number of samples that the algorithm detects wrong); $F_P$ means that no abnormal behavior is detected, and there is no abnormal behavior in the actual graph (the number of correct samples needed for the algorithm to detect errors); the recall rate ($R$) is the ratio of the number of abnormal behaviors detected to the total number of abnormal behaviors; the precision rate ($P$) is the ratio of the number of correctly detected abnormal behaviors to the total number of abnormal behaviors [33–35]. The average precision ($AP$) measures the accuracy of the model from the two aspects of precision and recall. It is a direct evaluation standard for model accuracy, and it can also be analyzed using the detection effect of a single category.

$$R = \frac{T_P}{T_P + F_N} \tag{14}$$

$$P = \frac{T_P}{T_P + F_P} \tag{15}$$

The abnormal behavior recognition results obtained according to Equations (14) and (15) are shown in Table 4.

**Table 4.** Evaluation indicators of behavior detection.

| Abnormal Behavior | *P*/% | *R*/% | *mAP*/% |
|-------------------|-------|-------|---------|
| Smoking           | 85.54 | 85.54 | 89.23   |
| Calling           | 75.76 | 87.36 | 87.35   |

## 5. Conclusions

An abnormal pilot behavior monitoring method based on the improved YOLO v4 algorithm was proposed. The method was verified by collecting abnormal behavior recognition datasets. The recognition rate was improved compared to the original basis. The CSPDarkent-53 framework was used to train the recognition model, which enhanced the method. The robustness of the training model was 85.54% for docking calls and smoking recognition. This method expands the training set through data augmentation, thereby achieving high-accuracy recognition with less training data. The algorithm performance needs to be further improved in later research. The next step is to explore the implantation of the algorithm into the camera terminal for practical applications.

The deep learning algorithm based on the improved YOLOv4 abnormal driving behavior monitoring algorithm can effectively identify the abnormal driving behavior of pilots. The attention mechanisms (CAB and SAB) were introduced to enhance the model's perception in channels and spaces. The image semantic features are extracted based on the

deep neural network structure, and the image and video recognition and classification of the pilots' driving behavior are then completed.

In the next step, we will continue to improve the network so that the network is not limited to feature extraction in the spatial domain, and we will also add some information in the time domain so as to further improve the generalization ability of the model.

**Author Contributions:** Conceptualization, N.C. and Y.S.; formal analysis, investigation, writing of the original draft, N.C. and Y.M. All authors have read and agreed to the published version of the manuscript.

**Funding:** This research was funded by the National Natural Science Foundation of China, grant number U2033202; the Key R&D Program of the Sichuan Provincial Department of Science and Technology (2022YFG0213); and the Safety Capability Fund Project of the Civil Aviation Administration of China (ASSA2022/17).

**Data Availability Statement:** The data used to support the findings of this study are included within the article.

**Acknowledgments:** Written informed consent has been obtained from the patients to publish this paper.

**Conflicts of Interest:** The authors declare no conflict of interest.

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
