# Peer review of "Abnormal Cockpit Pilot Driving Behavior Detection Using YOLOv4 Fused Attention Mechanism"

_electronics, doi:10.3390/electronics11162538_

Round 1
Reviewer 1 Report
Literature survey is not adequate. Moreover, literature cited in the paper are out of date. Many new papers on machine learning/deep learning algorithms are not included. Ref 23-25 are not cited in the article.
Introduction can be re framed including the objectives of the article. Article should be proofread by Native English speaker.
Section 3.3. Attention Mechanism Fusion should be explained well.
Figure 4. must be drastically improved to help the reading. I suggest authors to change and put high resolution images.
Explain environment settings to perform simulations & implementations.
Please have a section on Discussion after section 4.2. and have some more numerical analysis from experiments, such as accuracy.
Incorporate the comparison graph and tabulate the parameters with results data.
Author Response
Dear Reviewer and Editor,
Thank you very much for your letter and the comments from the key reader about our paper submitted to Electronics (electronics-1859203). Overall the comments have been fair, encouraging and constructive. So, we have learned much from them.
After carefully studying the reviewers' comments and advice, we have made corresponding changes to all the paper. We resubmit the revised manuscript as well as a list of changes.
If you have any question about this paper, please don’t hesitate to let me know. Email account is chennongtian@hotmail.com, and Tel is +86-838-5183621.
Sincerely!
Nongtian CHEN
Response to Reviewer #1:
We appreciate greatly the valuable suggestions from the reviewer. We have carefully considered the comments and revised the manuscript accordingly.
- Literature survey is not adequate. Moreover, literature cited in the paper are out of date. Many new papers on machine learning/deep learning algorithms are not included. Ref 23-25 are not cited in the article.
Answer: Thank you for reviewing the manuscript. Listen and absorb the suggestion, this article has added with the Ref 23-25 (317)and many new papers on machine learning/deep learning algorithms. After the revision, it is marked in red. Please review and verify.
- Introduction can be re framed including the objectives of the article. Article should be proofread by Native English speaker.
Answer: Thank you for reviewing the manuscript. Listen and absorb the suggestion, the introduction part has been processed, refactored and corrected, the paper has been language-polished, and the revised part has been marked in red. Please review and verify.
Section 3.3. Attention Mechanism Fusion should be explained well.
Answer: Thank you for reviewing the manuscript. Listen and absorb the suggestion,
the article has been modified (259-264).It is marked in red. Please review and verify.
- Figure 4. must be drastically improved to help the reading. I suggest authors to change and put high resolution images
Answer: Thank you for reviewing the manuscript. Listen and absorb the suggestion, The photos are taken from the video, and the clarity of the picture has been improved.
- Explain environment settings to perform simulations & implementations.
Answer: Thank you for reviewing the manuscript. Listen and absorb the suggestion, the article has been modified (274-302).It is marked in red. Please review and verify.
- Please have a section on Discussion after section 4.2. and have some more numerical analysis from experiments, such as accuracy.
Answer: Thank you for reviewing the manuscript. Listen and absorb the suggestion, the article has been modified (340-343).It is marked in red. Please review and verify.
- Incorporate the comparison graph and tabulate the parameters with results data.
Answer: Thank you for reviewing the manuscript. Listen and absorb the suggestion, the article has been modified.It is marked in red. Please review and verify.

Reviewer 2 Report
In this article, the authors proposed a method to detect abnormal behavior of the pilots in cockpit. They integrated Yolo v4 and attention mechanism to effectively detect the abnormal behavior of the pilots. The presented idea worthy therefore I recommend some major suggestions to enhance the quality of the paper.
1. In the abstract limitation of the existing approaches are not highlighted and motivation of the proposed approach is not given. I recommend revising the abstract by adding a sentence, covering limitation of existing systems and motivation for the proposed system.
2. The introduction section doesn’t follow the standard rules of manuscript writing. The authors need to revise the introduction section by incorporating the following points:
a. Motivation: general (why the problem is important)
b. Problem statement
c. Current challenges
3. Literature review section is not well-structured and contains coverage of old literature. For instance, you can see the papers.
a) https://www.sciencedirect.com/science/article/pii/S0167739X21004295
b) https://www.sciencedirect.com/science/article/pii/S0950705122006761
c) https://arxiv.org/abs/2101.01073,
d) https://link.springer.com/article/10.1007/s11042-020-09406-3
e) https://www.mdpi.com/1424-8220/21/8/2811
f) https://ieeexplore.ieee.org/abstract/document/9207595
4. The author needs to highlight their main contributions in bullets not in paragraph and also revise the contributions.
5. The authors should add the implementation detail of the proposed model which is currently lacking in this version.
6. Authors need to compare their method with recent existing techniques and clearly mention the performance evaluation criteria for proposed technique. I will encourage to include the system performance, time complexity, limitation of the system, and comparison with other state-of-the-art.
7. Authors should the deeply revised conclusion section and discuss the limitations of the proposed method and their possible solutions in the future work.
Author Response
Dear Reviewer and Editor,
Thank you very much for your letter and the comments from the key reader about our paper submitted to Electronics (electronics-1859203). Overall the comments have been fair, encouraging and constructive. So, we have learned much from them.
After carefully studying the reviewers' comments and advice, we have made corresponding changes to all the paper. We resubmit the revised manuscript as well as a list of changes.
If you have any question about this paper, please don’t hesitate to let me know. Email account is chennongtian@hotmail.com, and Tel is +86-838-5183621.
Sincerely!
Nongtian CHEN
Response to Reviewer #2:
We appreciate greatly the valuable suggestions from the reviewer. We have carefully considered the comments and revised the manuscript accordingly.
- In the abstract limitation of the existing approaches are not highlighted and motivation of the proposed approach is not given. I recommend revising the abstract by adding a sentence, covering limitation of existing systems and motivation for the proposed system.
Answer: Thank you for reviewing the manuscript. Listen and absorb the suggestion, the article has been modified (13-14,24-32).After the revision, it is marked in red. Please review and verify.
- Introduction can be re framed including the objectives of the article. Article should be proofread by Native English speaker.The introduction section doesn’t follow the standard rules of manuscript writing. The authors need to revise the introduction section by incorporating the following points:
- Motivation: general (why the problem is important)
- Problem statement
- Current challenges
Answer: Thank you for reviewing the manuscript. Listen and absorb the suggestion, the article instruction has revised. After the revision, it is marked in red. Please review and verify.
- Literature review section is not well-structured and contains coverage of old literature. For instance, you can see the papers.
- a)https://www.sciencedirect.com/science/article/pii/S0167739X21004295
- b)https://www.sciencedirect.com/science/article/pii/S0950705122006761
- c)https://arxiv.org/abs/2101.01073,
- d)https://link.springer.com/article/10.1007/s11042-020-09406-3
- e)https://www.mdpi.com/1424-8220/21/8/2811
- f)https://ieeexplore.ieee.org/abstract/document/9207595
Answer: Thank you for reviewing the manuscript. Listen and absorb the suggestion, the article has revised,and as an important reference for the paper, please refer to the reference section of the paper. After the revision, it is marked in red. Please review and verify.(97-106,420-430)
- The author needs to highlight their main contributions in bullets not in paragraph and also revise the contributions.
Answer: Thank you for reviewing the manuscript. Listen and absorb the suggestion, in order to better highlight the innovation of the article, corrections and supplements were made in the conclusion part, and the content is as follows:” The deep learning algorithm based on the improved YOLO v4 abnormal driving behavior monitoring can effectively identify the abnormal driving behavior of pilots. Introduce attention mechanisms (CAB and SAB) to enhance the model's perception in channels and spaces. The image semantic features are extracted based on the deep neural network structure, and the image and video recognition and classification of the pilot's driving behavior are completed.” it is marked in red. Please review and verify.( 376-381)
- The authors should add the implementation detail of the proposed model which is currently lacking in this version.
Answer: Thank you for reviewing the manuscript. Listen and absorb the suggestion,
the article has been modified (259-264). It is marked in red. Please review and verify.
- Authors need to compare their method with recent existing techniques and clearly mention the performance evaluation criteria for proposed technique. I will encourage to include the system performance, time complexity, limitation of the system, and comparison with other state-of-the-art.
Answer: Thank you for reviewing the manuscript. Listen and absorb the suggestion, the article has been modified (340-343). It is marked in red. Please review and verify.
- Authors should the deeply revised conclusion section and discuss the limitations of the proposed method and their possible solutions in the future work.
Answer: Thank you for reviewing the manuscript. Listen and absorb the suggestion, the article has been modified (382-385).It is marked in red. Please review and verify.

Round 2
Reviewer 1 Report
Article shall be accepted.
Reviewer 2 Report
All of my comments are addressed well, I recommend accept decision for this paper.